# ON THE DISCONNECT BETWEEN THEORY AND PRACTICE OF OVERPARAMETRIZED NEURAL NETWORKS

## ABSTRACT

The infinite-width limit of neural networks (NNs) has garnered significant attention as a theoretical framework for analyzing the behavior of large-scale, overparametrized networks. By approaching infinite width, NNs effectively converge to a linear model with features characterized by the neural tangent kernel (NTK). This establishes a connection between NNs and kernel methods, the latter of which are well understood. Based on this link, theoretical benefits and algorithmic improvements have been hypothesized and empirically demonstrated in synthetic architectures. These advantages include faster optimization, reliable uncertainty quantification and improved continual learning. However, current results quantifying the rate of convergence to the kernel regime suggest that exploiting these benefits requires architectures that are orders of magnitude wider than they are deep. This assumption raises concerns that practically relevant architectures do not exhibit behavior as predicted via the NTK. In this work, we empirically investigate whether the limiting regime either describes the behavior of large-width architectures used in practice or is informative for algorithmic improvements. Our empirical results demonstrate that this is *not* the case in optimization, uncertainty quantification or continual learning. This observed disconnect between theory and practice calls into question the practical relevance of the infinite-width limit.

## 1 INTRODUCTION

The behavior of large-scale, overparametrized neural networks (NNs) has for a long time been poorly understood theoretically. This is in stark contrast to their state-of-the-art performance on tasks like image classification (He et al., 2016; Zagoruyko & Komodakis, 2016), natural language processing (Devlin et al., 2019; Sun et al., 2019), as well as generative and sequence modeling (Brown et al., 2020; Touvron et al., 2023). The seminal work of Jacot et al. (2018) established a link between the evolution of NNs during training and kernel methods by considering networks with infinite width. In this limit, NNs effectively converge to linear models with fixed features such that their predictions are equivalent to those made by a Gaussian process (GP) model using the neural tangent kernel (NTK). Kernel methods and GPs are theoretically well-understood (Rasmussen & Williams, 2005). Consequently, this finding has led to a flurry of research interest in the NTK with the hope of an improved understanding of the behavior of NNs (e.g. Du et al., 2019; Zhou et al., 2020; Bowman & Montúfar, 2022b; Mirzadeh et al., 2022).

Kernel methods enjoy several benefits which are desirable for NNs. First, training a linear model or kernel regressor requires solving a quadratic optimization problem, which reduces to solving a linear system with the kernel matrix evaluated pairwise at the training data (Rasmussen & Williams, 2005). Conceptually this simplifies training significantly as the well-studied machinery of convex optimization and numerical linear algebra can be exploited. This is in contrast to the challenges of large-scale stochastic optimization, which compared to the convex setting suffers from slow convergence, requires manual tuning, and choosing an optimizer from a long list of available methods (Schmidt et al., 2021). Second, via the connection to GPs in the case of regression, uncertainty can be quantified via the posterior covariance defined through the NTK. As for prediction, uncertainty quantification then reduces to well-studied numerical methods (Rasmussen & Williams, 2005), unlike Bayesian NNs which generally suffer from similar issues as optimization (Zhang et al., 2020; Kristiadi et al., 2022a). Third, data often becomes available continually and we want to incorporate it into the model rather than retrain from scratch. This *continual learning* setting in practice often

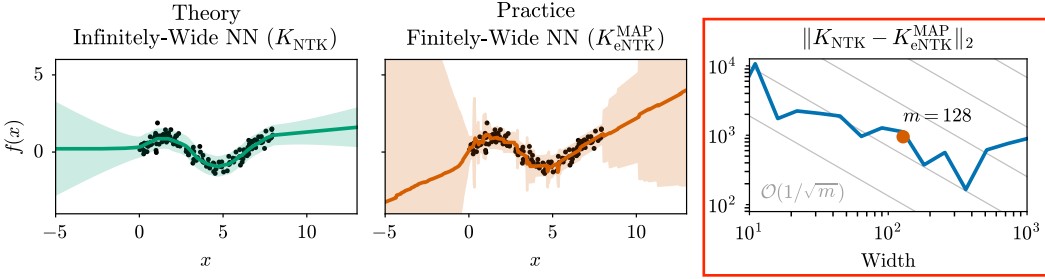

**Figure 1:** *Infinitely-wide NN in theory and its finite-width approximation in practice.*[1] The two models make very different predictions about the data-generating latent function, suggesting that the finite-width NN with a commonly selected architecture for real-world regression ($L = 3, m = 128, P = 21M$, Li et al., 2023) is not well-described by the kernel regime. Increasing the width by an order of magnitude does not significantly improve the approximation via the empirical NTK.

leads to a drop in performance on previous tasks, known as *catastrophic forgetting* (McCloskey & Cohen, 1989; Goodfellow et al., 2013). It has been observed that large-scale overparametrized networks forget less catastrophically (Ramasesh et al., 2022; Mirzadeh et al., 2022) and this has been hypothesized to be a consequence of these NNs behaving according to the NTK regime. If this were indeed the reason, worst-case forgetting could be predicted theoretically (Evron et al., 2022; 2023; Goldfarb & Hand, 2023) and mitigated algorithmically (Bennani et al., 2020; Doan et al., 2021).

Given the advantageous network properties and algorithmic implications in terms of training, uncertainty quantification, and continual learning close to the kernel regime, the question becomes *when* a network architecture satisfies the necessary assumptions. Loosely speaking, most theoretical results on NN convergence to a kernel regressor give rates of the form $\tilde{O}(1/\sqrt{m})$ in the (minimum) width of the hidden layers $m$ (Du et al., 2019; Lee et al., 2019; Bowman & Montúfar, 2022b). However, this asymptotic notation suppresses a dependence on the *network depth* $L$, which is generally at least *polynomial* (Arora et al., 2019) or even *exponential* (Bowman & Montúfar, 2022b). Even for quite shallow networks, this requires layer widths that are orders of magnitude larger than any of the common architectures (such as WideResNets). Figure 1 illustrates, that even shallow networks, if not sufficiently wide, can behave very differently from their infinite-width Gaussian process limit. This prompts the important question of whether assumptions based on the kernel regime, and methods derived thereof, apply to deep architectures that are used in practice.

**Contribution** In this work, we consider three areas for which the neural tangent kernel perspective on neural networks promises to be informative: optimization, uncertainty quantification and continual learning. We empirically evaluate whether the infinite-width regime either describes the behavior of large-width architectures used in practice or is useful for algorithm design. We find that in all three domains, assumptions based on NTK theory do *not* translate to predictable phenomena or improved performance. This disconnect between theory and practice challenges the significance of overparametrization theory when applied to common architectures. We hope our negative findings can serve as a call to action for theory development and as a cautionary tale for algorithm design.

**Limitations** Our work studies architectures that are *currently* being used in practice. This does *not* mean that future architectures with large widths are not described well via the kernel regime. However, achieving competitive performance with wide NNs is a challenge, likely due to reduced representation learning (Pleiss & Cunningham, 2021; Zavatone-Veth et al., 2021; Noci et al., 2021; Coker et al., 2022), and the NTK does not predict the scaling laws of finite-width NNs well (Vyas et al., 2023). Finally, we do *not* claim that the methods we consider fail to be competitive in *any* setting, rather just that their motivation via the kernel regime is unsuitable for practical architecture choices and problems. They may work well on specific choices of models and datasets.

## 2 OVERPARAMETRIZATION THEORY: AN OVERVIEW

Let $f : \mathcal{X} \times \Theta \to \mathcal{Y}$ be a neural network (NN) with input space $\mathcal{X} \subseteq \mathbb{R}^D$, output space $\mathcal{Y} \subseteq \mathbb{R}^C$ and parameter space $\Theta \subseteq \mathbb{R}^P$. Assume we linearize $f$ around a parameter vector $\boldsymbol{\theta}_0 \in \Theta$, i.e.

$$f(\boldsymbol{x};\boldsymbol{\theta}) \approx f_{\text{lin}}(\boldsymbol{x};\boldsymbol{\theta}) := f(\boldsymbol{x};\boldsymbol{\theta}_0) + \boldsymbol{J}(\boldsymbol{x};\boldsymbol{\theta}_0)(\boldsymbol{\theta} - \boldsymbol{\theta}_0) \tag{1}$$

---

[1]Computed via the `neural-tangents` library (Novak et al., 2020).

where $\boldsymbol{J}(\boldsymbol{x}; \boldsymbol{\theta}_0) := (\partial f_{\boldsymbol{\theta}}(\boldsymbol{x})/\partial\boldsymbol{\theta})|_{\boldsymbol{\theta}=\boldsymbol{\theta}_0} \in \mathbb{R}^{C \times P}$. When $\boldsymbol{\theta}$ is close to $\boldsymbol{\theta}_0$, the linear model $f_{\text{lin}}(\boldsymbol{x}; \boldsymbol{\theta})$ with features defined by the network's Jacobian $\boldsymbol{J}(\boldsymbol{x}; \boldsymbol{\theta}_0)$ is a good approximation of $f(\boldsymbol{x}; \boldsymbol{\theta})$. Consider a fully connected neural network $f_{\text{MLP}}(\boldsymbol{x}; \boldsymbol{\theta}) = \boldsymbol{h}^L(\boldsymbol{x}^{L-1})$ defined recursively as

$$\boldsymbol{h}^\ell(\boldsymbol{x}^{\ell-1}) = \boldsymbol{W}^\ell \boldsymbol{x}^{\ell-1} + \boldsymbol{b}^\ell, \qquad \boldsymbol{x}^{\ell-1}(\boldsymbol{h}^{\ell-1}) = \varphi(\boldsymbol{h}^{\ell-1}), \qquad \boldsymbol{x}^0 = \boldsymbol{x} \qquad (2)$$

for $\ell = L, \ldots, 1$ with parameters $\boldsymbol{\theta} = \{\boldsymbol{W}^\ell = {}^1\!/\!{\sqrt{m_{\ell-1}}}\boldsymbol{V}^\ell\}_{\ell=1}^L \cup \{\boldsymbol{b}^\ell\}_{\ell=1}^L$, s.t. $\boldsymbol{V}_{ij}^\ell, \boldsymbol{b}_i^\ell \sim \mathcal{N}(0,1)$, layer widths $m_\ell$ and activation function $\varphi$.[2] Remarkably, Jacot et al. (2018) showed that for infinitely wide fully connected NNs, the parameters remain sufficiently close to their initialization $\boldsymbol{\theta}_0$ during training via gradient descent. This means we can (approximately) understand the properties of a wide NN $f_{\text{MLP}}(\boldsymbol{x}; \boldsymbol{\theta})$ by considering a much simpler-to-understand *linear* model with features defined by the Jacobian $\boldsymbol{J}(\boldsymbol{x}; \boldsymbol{\theta}_0)$ instead. Or more precisely, from a function space perspective, in the infinite width limit the behavior of a fully connected NN is described by the (deterministic) *neural tangent kernel* $K_{\text{NTK}}$, defined as the limit in probability of the *finite-width* or *empirical NTK*

$$K_{\text{eNTK}}^{\boldsymbol{\theta}}(\boldsymbol{x}, \boldsymbol{x}') := \boldsymbol{J}(\boldsymbol{x}; \boldsymbol{\theta})\boldsymbol{J}(\boldsymbol{x}'; \boldsymbol{\theta})^\top \xrightarrow{P} K_{\text{NTK}}(\boldsymbol{x}, \boldsymbol{x}') \quad \text{as } m_1, \ldots, m_L \to \infty.$$

This is known as the linear or *kernel regime*. In this regime, at initialization, the implicit prior over functions defined by the network is given by a Gaussian process $\mathcal{GP}(0, K_{\text{NTK}})$ with zero-mean and covariance function defined by the NTK. Further, the (continuous-time) training dynamics of the network can be described via the differential equation $\partial_t f(\boldsymbol{x}; \boldsymbol{\theta}_t) = -K_{\text{NTK}}(\boldsymbol{x}, \boldsymbol{X})\nabla_f \mathcal{L}(f(\boldsymbol{X}; \boldsymbol{\theta}_t))$ i.e. the optimization trajectory of $f(\boldsymbol{x}; \boldsymbol{\theta})$ is given by kernel gradient descent with respect to the loss function $\mathcal{L}$. In the case of square loss regression on a training dataset $(\boldsymbol{X}, \boldsymbol{y})$, this is a linear ODE, which in the limit of infinite training $t \to \infty$ admits a closed-form solution. The network prediction is equivalent to the mean function

$$\lim_{t \to \infty} f(\boldsymbol{x}; \boldsymbol{\theta}_t) = \mu_*(\boldsymbol{x}) := f(\boldsymbol{x}; \boldsymbol{\theta}_0) + K_{\text{NTK}}(\boldsymbol{x}, \boldsymbol{X})K_{\text{NTK}}(\boldsymbol{X}, \boldsymbol{X})^{-1}(\boldsymbol{y} - f(\boldsymbol{X}; \boldsymbol{\theta}_0)) \qquad (3)$$

of a GP posterior $\mathcal{GP}(\mu_*, K_*)$, resulting from conditioning the prior $\mathcal{GP}(0, K_{\text{NTK}})$ on observations $\boldsymbol{y} = f_*(\boldsymbol{X})$ from the latent function $f_*$ generating the data. These results for fully connected NNs have since been extended to nearly all architectures currently used in practice, such as CNNs, RNNs, and GNNs (Yang & Littwin, 2021).

**Implications for Training, Uncertainty Quantification and Continual Learning**  The connection to GP regression with the NTK demonstrates why the kernel regime is powerful as a theoretical framework. First, training a neural network simplifies to solving a linear system or equivalently a convex optimization problem (assuming $K_{\text{NTK}}$ is positive (semi-)definite) since

$$K_{\text{NTK}}(\boldsymbol{X}, \boldsymbol{X})^{-1}(\boldsymbol{y} - f(\boldsymbol{X}; \boldsymbol{\theta}_0)) = \arg\min_{\boldsymbol{v}} \tfrac{1}{2}\boldsymbol{v}^\top K_{\text{NTK}}(\boldsymbol{X}, \boldsymbol{X})\boldsymbol{v} - (\boldsymbol{y} - f(\boldsymbol{X}; \boldsymbol{\theta}_0))^\top \boldsymbol{v} \qquad (4)$$

which can be solved using well-understood, fast converging algorithms from numerical analysis (Nocedal & Wright, 2006). This is in contrast to the challenges of stochastic, non-convex optimization (Schmidt et al., 2021). Second, one often cited limitation of NNs is their lack of uncertainty quantification (Hein et al., 2019; Kristiadi et al., 2022b;a). The connection to the posterior mean of a GP in the kernel regime when training to convergence (3) provides a strategy for Bayesian deep learning (Lee et al., 2018; Khan et al., 2019), by using the posterior covariance function

$$K_*(\boldsymbol{x}, \boldsymbol{x}') := K_{\text{NTK}}(\boldsymbol{x}, \boldsymbol{x}') - K_{\text{NTK}}(\boldsymbol{x}, \boldsymbol{X})K_{\text{NTK}}(\boldsymbol{X}, \boldsymbol{X})^{-1}K_{\text{NTK}}(\boldsymbol{X}, \boldsymbol{x}') \qquad (5)$$

to quantify uncertainty. Finally, in a continual learning problem, the similarity between tasks in the kernel regime is measured via the NTK, which in turn describes the amount of catastrophic forgetting when training on new tasks (Doan et al., 2021).

**Convergence to the Kernel Regime**  When should we expect a neural network's behavior to be well-described by the NTK? We can characterize how quickly a network approaches the kernel regime as a function of its (minimum) width $m = \min_{\ell \in \{1, \ldots, L-1\}} m_\ell$. The typical rate of convergence of the finite-width NTK at initialization to the NTK is

$$|K_{\text{eNTK}}^{\boldsymbol{\theta}_0}(\boldsymbol{x}, \boldsymbol{x}') - K_{\text{NTK}}(\boldsymbol{x}, \boldsymbol{x}')| = \tilde{O}({}^1\!/\!{\sqrt{m}}) \qquad (6)$$

---

[2]This normalized form of the weight matrices is known as the *NTK parametrization* (NTP).

either pointwise (Du et al., 2019; Arora et al., 2019; Huang & Yau, 2020) or uniform (Buchanan et al., 2021; Bowman & Montúfar, 2022a;b) with high probability. These results assume an *over-parametrized* NN with width $m = \Omega(\text{poly}(N))$ significantly exceeding the number of training datapoints $N$.[3] Note that the asymptotic notation in (6) suppresses a dependence on the (constant) depth $L$ of the NN. This dependence of the width on the depth is *polynomial* (e.g. $m = \Omega(L^6)$ in Arora et al., 2019) or even *exponential* (Bowman & Montúfar, 2022b), which suggests that to approach the kernel regime, a large network width is required already at moderate depth.

For most architectures, an analytical/efficient-to-evaluate expression for the NTK is not known.[4] Therefore in practice, the finite-width NTK $K^{\boldsymbol{\theta}}_{\text{eNTK}} \approx K_{\text{NTK}}$ is used as an approximation. However as Fig. 1 illustrates, the prediction of a finite-width NN and the associated uncertainty given by the empirical NTK can be very different from the network's theoretical infinite-width limit. Therefore, making assumptions based on the kernel regime can potentially be misleading.

# 3 CONNECTING THEORY AND PRACTICE

To empirically study whether the predictions from the kernel regime about the behavior of over-parametrized NNs are reproducible in architectures used in practice, we take a closer look at training, uncertainty quantification and continual learning. For each of these, the kernel regime either makes predictions about the behavior of the network, motivates certain algorithms, or both.

## 3.1 TRAINING: CONDITIONS FOR FAST CONVERGENCE OF SECOND-ORDER OPTIMIZATION

The empirical risk is typically a convex function of the network output, but generally non-convex in the network's parameters. In addition, stochastic approximations are often necessary due to memory constraints. However, for a NN close to the kernel regime, informally, the loss landscape becomes more convex, since the network approaches a linear model. In fact, for square loss the problem becomes quadratic, see (4). Using this intuition about the kernel regime, Du et al. (2019) have shown that gradient descent, a first-order method, can achieve zero training loss in spite of non-convexity. First-order methods, such as SGD and ADAM, are state-of-the-art for deep learning optimization (Schmidt et al., 2021). This is in contrast to "classical" convex optimization, in which second-order methods are favored due to their fast convergence (e.g. Nesterov, 2008; 2021). If NNs in practice are described well theoretically via the kernel regime, this may seem like a missed opportunity, since the near-convexity of the problem would suggest second-order optimizers to be an attractive choice.

There are multiple challenges for making second-order methods practical. They are less stable under noise—predominant in deep learning where data is fed in mini-batches—have higher per-iteration costs, and are often more complex to implement efficiently. Here, we investigate whether the theoretical argument in favor of second-order methods applies to real-world networks in that they are sufficiently close to the kernel regime. We exclude the aforementioned additional challenges, which amount to an entire research field (e.g. Martens & Grosse, 2015; Gupta et al., 2018; Ren & Goldfarb, 2021), since in the overparametrization regime approximate second-order methods that overcome these challenges can be shown to exhibit similar behavior than their deterministic counterparts (Karakida & Osawa, 2020).

**Fast Convergence of NGD in the Kernel Regime** To empirically test whether (practical) NNs can be efficiently optimized via second-order optimization as predicted by theory in the infinite-width limit, we consider natural gradient descent (NGD). Zhang et al. (2019) studied the convergence behavior of NGD theoretically. They give conditions for fast convergence of finite-step NGD, which extend to approximate NGD methods such as KFAC (Martens & Grosse, 2015). For simplicity, we focus on the special case of NNs with scalar-valued output and mean-squared error. Consider a network $f(\boldsymbol{\theta}, \boldsymbol{x}) := f(\boldsymbol{x}; \boldsymbol{\theta})$ with flattened parameters $\boldsymbol{\theta} \in \mathbb{R}^P$. For a dataset $\{(\boldsymbol{x}_n, y_n)\}_{n=1}^N$, we minimize the empirical risk $\mathcal{L}(\boldsymbol{\theta}) = 1/2N \|\boldsymbol{f} - \boldsymbol{y}\|_2^2$ where $\boldsymbol{f}(t) := (f(\boldsymbol{\theta}(t), \boldsymbol{x}_1) \ldots f(\boldsymbol{\theta}(t), \boldsymbol{x}_N))^\top \in \mathbb{R}^N$ and $\boldsymbol{y} := (y_1 \ldots y_N)^\top$. Zhang et al. (2019) describe two conditions to ensure fast convergence of finite-step NGD to a global minimum which apply to *arbitrary* architectures:

---

[3]Note, that Bowman & Montúfar (2022b) relax this requirement to $m \approx N$ via the use of a stopping time.

[4]A fully connected neural network with ReLU activations being a notable exception (Lee et al., 2018).

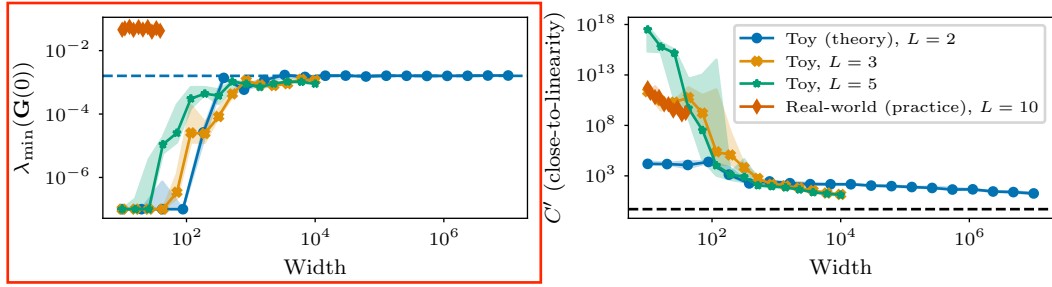

**Figure 2:** *Conditions for fast convergence of NGD for different NN widths and problems* (dots represent medians and shaded areas represent the 25/75-quantiles over three independent initializations and five parameter perturbations). The problems range from shallow (*theory*) ReLU MLP and depth-3 and -5 ReLU MLPs on a synthetic regression task to WideResNets on a sub-set of CIFAR-10 with $N = 400$ (*practice*). *Left:* Relatively small widths are sufficient to satisfy the Gram matrix condition (7) (dashed line corresponds to the theory prediction of the limiting eigenvalue $\lambda_0$ for the toy problem). *Right:* None of the NNs achieve the required Jacobian stability (horizontal dashed line at $1/2$) from (8) for any width both for synthetic and benchmark data.

1. *Full row-rank of the network Jacobian $\boldsymbol{J}(\boldsymbol{X}; \boldsymbol{\theta}(0)) \in \mathbb{R}^{N \times P}$ at initialization*, implying

$$\lambda_{\min}(\boldsymbol{G}(0)) > 0, \tag{7}$$

where $\boldsymbol{G}(0) = K_{\mathrm{eNTK}}^{\boldsymbol{\theta}(0)}(\boldsymbol{X}, \boldsymbol{X})$ and restricting the trajectory to be close to initialization.[5]

2. *Stable Jacobian*, $\exists 0 \leq C \leq 1/2$ such that $\forall \boldsymbol{\theta} : \|\boldsymbol{\theta} - \boldsymbol{\theta}(0)\|_2 \leq \rho := 3\|\boldsymbol{y} - \boldsymbol{f}(0)\|_2 / \sqrt{\lambda_{\min}(\boldsymbol{G}(0))}$

$$\|\boldsymbol{J}(\boldsymbol{\theta}) - \boldsymbol{J}(0)\|_2 \leq \frac{C}{3}\sqrt{\lambda_{\min}(\boldsymbol{G}(0))}. \tag{8}$$

The smaller $C$, the 'closer' to linear the network is to initialization, with equality for $C = 0$.

We can evaluate both conditions in a scalable, matrix-free fashion using standard functions of automatic differentiation frameworks (see Section A.1). As a proxy for (8), we compute $C'(\boldsymbol{\theta}) := 3\|\boldsymbol{J}(\boldsymbol{\theta}) - \boldsymbol{J}(0)\|_2 / \sqrt{\lambda_{\min}(\boldsymbol{G}(0))}$ with $\boldsymbol{\theta}$ drawn uniformly from a sphere with radius $\rho$. If $C'(\boldsymbol{\theta}) > 1/2$ for any $\boldsymbol{\theta}$, then $C \notin [0; 1/2]$ and the network violates the stable Jacobian condition. Figures 2 and 3 summarize our findings which we now discuss in more detail.

**Shallow ReLU Net + Synthetic Data** We start with a synthetic problem for which Zhang et al. (2019) give theoretical guarantees. We generate a regression problem with $N = 16$ by i.i.d. drawing $\boldsymbol{x}_n \in \mathbb{R}^2 \sim \mathcal{U}([0; 1]^2)$, $\epsilon_n \in \mathbb{R} \sim \mathcal{N}(0, 1)$, and setting $y_n = \sin(2\pi([\boldsymbol{x}_n]_1 + [\boldsymbol{x}_n]_2)) + 0.1\epsilon_n$. Our model is a shallow two-layer ReLU net $f(\boldsymbol{x}, \boldsymbol{\theta}) = 1/\sqrt{m}\boldsymbol{W}^{(2)}\mathrm{ReLU}(\boldsymbol{W}^{(1)}\boldsymbol{x})$ where $\boldsymbol{W}^{(1)} \in \mathbb{R}^{m \times 2} \sim \mathcal{N}(\boldsymbol{0}, \nu^2 \boldsymbol{I})$ with $\nu = 1$, $\boldsymbol{W}^{(2)} \in \mathbb{R}^{1 \times m} \sim \mathcal{U}(\{-1, 1\}^m)$. Only $\boldsymbol{W}^{(1)}$ is trainable and each input is normalized in the pre-processing stage, $\boldsymbol{x}_n \leftarrow \boldsymbol{x}_n / \|\boldsymbol{x}_n\|_2$, to satisfy the theoretical assumptions. In this setting, Zhang et al. (2019) show that $m = \Omega(N^4/\nu^2\lambda_0^4\delta^3)$ is required for fast convergence of NGD with probability at least $1 - \delta$ and achieves an improvement of $\mathcal{O}(\lambda_0/N)$ over GD.[6] The Jacobian has full row-rank with high probability for $m = \Omega(N\log(N/\delta)/\lambda_0)$ and we empirically observe a sharp increase in $\lambda_{\min}(\boldsymbol{G}(0))$ at relatively low widths (around $m = 500$) in Fig. 2.

However, the Jacobian stabilizes with $\|\boldsymbol{J}(\boldsymbol{\theta}) - \boldsymbol{J}(0)\|_2 = \mathcal{O}(m^{-1/6})$, and even for extreme widths (up to $10^7$) we observe that $C' > 1/2$, and therefore $C > 1/2$.

**Deep ReLU Net + Synthetic Data** Next, we move away from the kernel regime by adding depth to the architecture while keeping the same synthetic data and pre-processing. We use two fully connected NNs, as defined in (2), with $L \in \{3, 5\}$ layers of equal width and ReLU activations. For these models, scaling to large

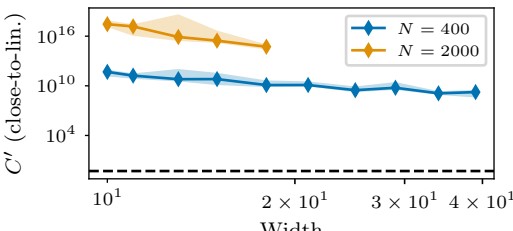

**Figure 3:** More data *decreases* the Jacobian stability for WideResNet on a subset of CIFAR-10.

---

[5]Note, that this implicitly assumes $P \geq N$, i.e. overparametrization.

[6]Here $\lambda_0 = \lambda_{\min}(K_{\mathrm{NTK}}(\boldsymbol{X}, \boldsymbol{X}))$ is the minimum eigenvalue of the NTK from Du et al. (2019).

widths is more expensive than for the shallow case, as the NN's parameters grow quadratically in $m$. For both depths, we observe a sharp transition in $\lambda_{\min}(\boldsymbol{G}(0))$ at relatively small widths (around $m = 500$) that are computationally accessible. In the close-to-linearity measure, we can see that depth increases non-linearity. While we observe a similar sharp transition in $C'$ to smaller values around $m = 500$ for both depths, its values remain well above $1/2$.

**CNN + Benchmark Data** Finally, we investigate a practical architecture (WideResNet, depth $L = 10$) on CIFAR-10. We convert the classification task into a regression problem for class indices and use a subset of the data. We rely on the implementation of Kuen (2017) and use its built-in widening factor, which scales the channels of the intermediate features within a block, as a measure for the network's width $m$. In contrast to the previous cases, this net's Jacobian has a full row rank even for small widths. However, for larger widths attainable within our compute budget, $C'$ remains many orders of magnitude above $1/2$. And the stability further deteriorates when using more data (Fig. 3).

**Summary:** In the kernel regime, NGD has favorable convergence over GD in theory. However, empirically we find that the necessary conditions consistently do *not* hold throughout problem scales—even for a shallow network with theoretical guarantees.

## 3.2 Uncertainty Quantification: Neural Bandits

In sequential decision-making problems, not only predictive accuracy of a model is important, but crucially also accurate uncertainty quantification (Lattimore & Szepesvári, 2020; Garnett, 2023). Recently, the connection between infinitely wide NNs and GPs has been exploited to design neural bandit algorithms, whose guarantees rely on the assumption that the surrogate model $f_{\boldsymbol{\theta}_t}$ is sufficiently close to the kernel regime (Zhou et al., 2020; Zhang et al., 2021; Kassraie & Krause, 2022; Nguyen-Tang et al., 2022). We empirically test whether this assumption holds in practice.

**Neural Contextual Bandits via the Kernel Regime** Our goal is to sequentially take optimal actions with regard to an unknown time-varying reward function $r_t(a_t, \boldsymbol{x}_t) \in \mathbb{R}$ which depends on an action-context pair $(a_t, \boldsymbol{x}_t) \in \{1, \ldots, K\} \times \mathbb{R}^n$ where $t = 1, \ldots, T$. To do so, we learn a surrogate $f_{\boldsymbol{\theta}_t}(a_t, \boldsymbol{x}_t) \approx r_t(a_t, \boldsymbol{x}_t)$ approximating the reward from past data $\mathcal{D}_t = \{(a_{t'}, \boldsymbol{x}_{t'}), r_{t'}(a_{t'}, \boldsymbol{x}_{t'})\}_{t'=1}^{t-1}$. An action $\widetilde{a}_t = \arg\max_{a_t} u(a_t, \boldsymbol{x}_t)$ is then selected based on a *utility function* $u$ which generally depends on both the prediction and uncertainty of the neural surrogate for the reward. Overall we want to minimize the *cumulative regret* $R(T) = \sum_{t=1}^{T} r_t(\widetilde{a}_t, \boldsymbol{x}_t) - r_t(a_t^*, \boldsymbol{x}_t)$, where $a_t^*$ are the optimal actions, and the reward $r_t(\widetilde{a}_t, \boldsymbol{x}_t)$ is only observable once an action $\widetilde{a}_t$ is taken. Here we use the popular UCB (Auer, 2002) utility function

$$u(a_t, \boldsymbol{x}_t) = f_{\boldsymbol{\theta}_t}(a_t, \boldsymbol{x}_t) + \gamma_t \sqrt{\operatorname{var} f_{\boldsymbol{\theta}_t}(a_t, \boldsymbol{x}_t)}, \tag{9}$$

where $\gamma_t > 0$ controls the exploration-exploitation tradeoff. Due to the importance of uncertainty quantification in the selection of an action based on $u(a_t, \boldsymbol{x}_t)$, GPs have been used extensively as surrogates (Krause & Ong, 2011). Here, we consider neural surrogates instead, which quantify uncertainty via the empirical NTK at a MAP estimate $\boldsymbol{\theta}_t$. This can be thought of as a finite-width approximation to the limiting GP in the kernel regime, or equivalently from a weight space view as a linearized Laplace approximation (LLA, MacKay, 1992; Khan et al., 2019; Immer et al., 2021b; Daxberger et al., 2021; Kristiadi et al., 2023b).

**Exploration-Exploitation Trade-off** The parameter $\gamma_t$ in (9) controlling the exploration-exploitation trade-off strongly impacts the cumulative regret, making its choice an important problem in practice. Recent works prove (near-)optimal regret bounds for the neural bandit setting by choosing $\gamma_t$ based on the kernel regime (Zhou et al., 2020; Kassraie & Krause, 2022). To approach the kernel regime, the convergence results discussed in Section 2 re-

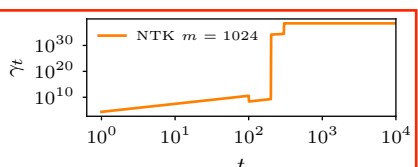

**Figure 4:** Setting $\gamma_t$ via NTK theory results in overexploration in practice.

quire the width of the network $m$ to be polynomial in the depth $L$ and number of training data $N = t - 1$. This poses the question whether the proposed choice of $\gamma_t$ is useful in practice. Here, we consider the NeuralUCB algorithm proposed by Zhou et al. (2020), where the exploration parameter $\gamma_t = \tilde{O}(\operatorname{poly}(1/\sqrt{m}, L, t))$. We find that even for shallow NNs ($L = 3$), $\gamma_t$ rapidly grows very large (see Fig. 4), which by (9) results in essentially no exploitation, only exploration. This suggests that for $\gamma_t$ to achieve a non-vacuous value, $m$ must be potentially unfeasibly large.

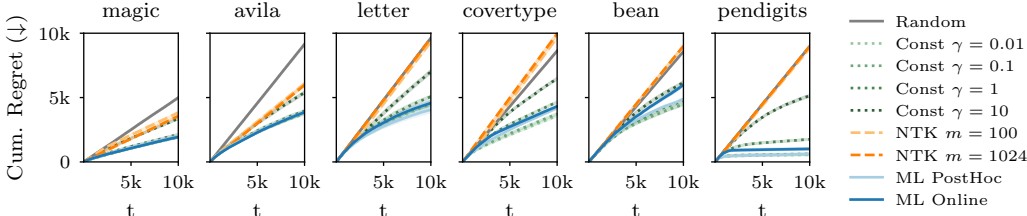

**Figure 5:** *Cumulative regret of neural bandits with different degrees of exploration* $(\gamma_t)_t$ *on benchmark data. Setting the exploration parameter* $\gamma_t$ *via NTK theory yields second-worst performance after random search. Constant exploration achieves the best results but the optimal* $\gamma_t \equiv \gamma = 10^{-2}$ *is a-priori unknown. Online marginal-likelihood (ML) calibration performs near-optimally.*

**Experiment Setup** We empirically test whether the assumptions based on the kernel regime in the neural bandit setting result in good performance in practice for realistic architectural choices. We use standard contextual bandit benchmark problems, based on UCI classification datasets (Zhou et al., 2020; Zhang et al., 2021; Gupta et al., 2022) (see Section A.2). We compare (i) a *random* baseline policy and various neural UCB baselines, (ii) the UCB policy with *constant* exploration parameter $\gamma_t \equiv \gamma \in \{0.01, 0.1, 1, 10\}$ as for simplicity often used in practice, (iii) the UCB policy where $\gamma_t$ is set via the NTK theory with widths $m \in \{100, 1024\}$ (Zhou et al., 2020), and (iv) setting $\gamma_t \equiv 1$, but leveraging the connection between the (empirical) NTK and the LLA in Bayesian deep learning (Immer et al., 2021b) to learn a prior precision hyperparameter via marginal likelihood both *post-hoc* and *online* (Immer et al., 2021a).[7]

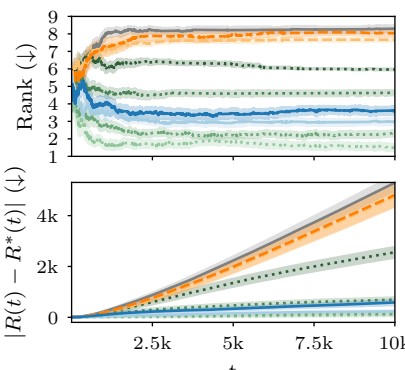

**Figure 6:** Ranking of methods for setting $\gamma_t$ w.r.t. cumulative regret and abs. difference to optimal regret. Averaged over datasets and random seeds.

**Experiment Results** The results of our experiment are given in Fig. 5, which shows the cumulative regret $R(t)$ over time. Perhaps frustratingly, the NTK-based policy performs poorly, oftentimes no better than the random baseline, with an order of magnitude larger width having no discernable effect. This is likely explained by the overexploration problem discussed previously. Therefore, in this setting, relying on assumptions based on the kernel regime results in a poorly performing algorithm in practice. In fact, Zhou et al. (2020) set $\gamma_t$ to be constant in their experiments instead of according to the proposed (near-)optimal value based on NTK theory. This disconnect between NTK theory and practice can also be observed for other utility functions such as expected improvement (Gupta et al., 2022) and Thompson sampling (Zhang et al., 2021). We find that setting $\gamma_t \equiv \gamma$ to a value with a well-chosen $\gamma$ performs best in our experiments (see also Fig. 6 top) However, the optimal value of $\gamma$ is *unknown a-priori* and can only be obtained via grid search. This can be problematic in a real-world setting, where a sufficiently large, representative, validation set may not be available, and multiple experiments may not be possible prior to running the "real" experiment—it defeats the spirit of *online* learning. With that in mind, the marginal-likelihood-based choice of $\gamma_t$ both post-hoc *and* online perform well in terms of their cumulative regret. While using grid search provides better results in terms of rank (Fig. 6 top), the difference in terms of the cumulative regret $R(t)$ is small for all $t$—see Fig. 6 bottom. The minimal difference in cumulative regret between the marginal-likelihood-based strategies and the best strategy suggests that learning a good exploration-exploitation trade-off is possible, but likely *not* via an algorithm motivated via the kernel regime.

**Summary:** Avoid setting the exploration parameter in eNTK-based neural bandits via the NTK theory. Instead, use the toolbox of the Laplace approximation to optimize the scale of the posterior variance via evidence maximization.

---

[7]Computing the evidence in the LLA setting incurs no overhead since the LA itself is an approximation of *both* the posterior $p(\boldsymbol{\theta} \mid \mathcal{D}_t)$ *and* the marginal likelihood $p(\mathcal{D}_t) = \int p(\mathcal{D}_t \mid \boldsymbol{\theta}) \, p(\boldsymbol{\theta}) \, d\boldsymbol{\theta}$ (MacKay, 1992).

### 3.3 CONTINUAL LEARNING: CATASTROPHIC FORGETTING

In many applications, such as robotics, NNs need to be trained continually on new *tasks*, given by a sequence of training datasets. The primary challenge in *continual learning* (Thrun & Mitchell, 1995; Parisi et al., 2019) is to train on new tasks without a significant loss of performance on previous tasks, known as *catastrophic forgetting* (McCloskey & Cohen, 1989; Goodfellow et al., 2013).

**Catastrophic Forgetting in the Kernel Regime**  Assuming the NN is sufficiently wide to be in the linear regime, worst-case forgetting and the convergence to an offline solution—i.e. training on data from all tasks at once–can be described theoretically (Evron et al., 2022; 2023; Goldfarb & Hand, 2023). One way to algorithmically avoid forgetting is *orthogonal gradient descent* (OGD, Farajtabar et al., 2020), which projects gradients on new tasks such that model predictions on previous tasks change minimally. Bennani et al. (2020) show that, in the kernel regime, OGD provably avoids catastrophic forgetting on an arbitrary number of tasks (assuming infinite memory). Additionally, for SGD and OGD generalization bounds have been given, which are based on the task similarity with respect to the NTK (Bennani et al., 2020; Doan et al., 2021). Ramasesh et al. (2022) investigated catastrophic forgetting empirically in the pretraining paradigm and found that forgetting systematically decreases with scale of both model and pretraining dataset size. Mirzadeh et al. (2022) report that increasing the width of a neural network reduces catastrophic forgetting significantly as opposed to increasing the depth. The hypothesis for this is that as the model becomes wider, gradients across tasks become increasingly orthogonal, and training becomes "lazy", meaning the initial parameters change very little during training, consistent with the convergence of the empirical NTK at initialization to the NTK (6). This naturally leads to the question of whether increasing the width of networks that are used in practice is in fact a simple way to mitigate catastrophic forgetting.

**Experiment Setup**  To test whether the predictions about continual learning in the kernel regime apply in practice, we train increasingly wide NNs on a sequence of tasks. We train toy two-layer NNs with ReLU activations on the RotatedMNIST dataset, as well as WideResNets (Zagoruyko & Komodakis, 2016) with depth $L = 10$ on the SplitCIFAR10, SplitCIFAR100 and SplitTinyImageNet datasets. See Section A.3 for details. Our main goal is to study the effect of width on forgetting. Let $a_{t,i}$ denote test accuracy on task $i$ after training on task $t$. We compute the *average forgetting* $\phi_T = {}^1/_{T-1}\sum_{i=1}^{T-1}\max_{t\in\{1,\ldots,T-1\}}(a_{t,i} - a_{T,i})$, i.e. the average maximal accuracy difference during task-incremental training; the *average accuracy* $\bar{\alpha}_T = {}^1/_T\sum_{i=1}^{T}a_{T,i}$, i.e. the average accuracy across tasks after training on all tasks; and the *learning accuracy* $\bar{\alpha}_{\max} = {}^1/_T\sum_{i=1}^{T}a_{i,i}$, i.e. the average accuracy across tasks immediately after training on the current task.[8] To ascertain whether a network operates in the lazy training/kernel regime, we also track the relative distance in parameter space $d(\boldsymbol{w}_T, \boldsymbol{w}_0) = \|\boldsymbol{w}_T - \boldsymbol{w}_0\|_2/\|\boldsymbol{w}_0\|_2$ between the initial parameters $\boldsymbol{w}_0$ and the parameters $\boldsymbol{w}_T$ after training on all tasks.

**Experiment Results**  The results of our experiment are shown in Fig. 7. We find that for the shallow NN, average forgetting decreases monotonically with the network width. Further, the relative change in parameters $d(\boldsymbol{w}_T, \boldsymbol{w}_0)$ approaches zero, consistent with the lazy training hypothesis in the kernel regime. This seems to confirm observations by Mirzadeh et al. (2022) that wide neural networks forget less. However, for WideResNets we find that a crucial confounding factor is whether the network has been fully trained. NNs that are trained insufficiently show a decrease in forgetting as they become wider. But, this decrease is primarily due to lower learning accuracy, and thus a smaller gap between peak accuracy and minimum accuracy across tasks (see Section B.2). In contrast, training networks to high accuracy increases average forgetting since the peak performance across tasks increases. This can be explained by the fact that they are not actually operating in the kernel regime. The relative change of the weights during training remains large even as width increases beyond what is used in practice, meaning the networks are still adapting to unseen tasks.

**Summary:** Increasing the width of a practical NN architecture only avoids catastrophic forgetting if not trained to high accuracy per task. Since a smaller change in the weights of the network throughout training correlates with reduced forgetting, strategies that constrain parameter updates when training on new tasks (e.g. Kirkpatrick et al., 2017) or along directions which minimally change performance on previous tasks (e.g. OGD) promise to be more useful strategies in practice than increasing network width.

---

[8]In practice, $\bar{\alpha}_{\max}$ almost always equals the average maximum accuracy per task, justifying the notation.

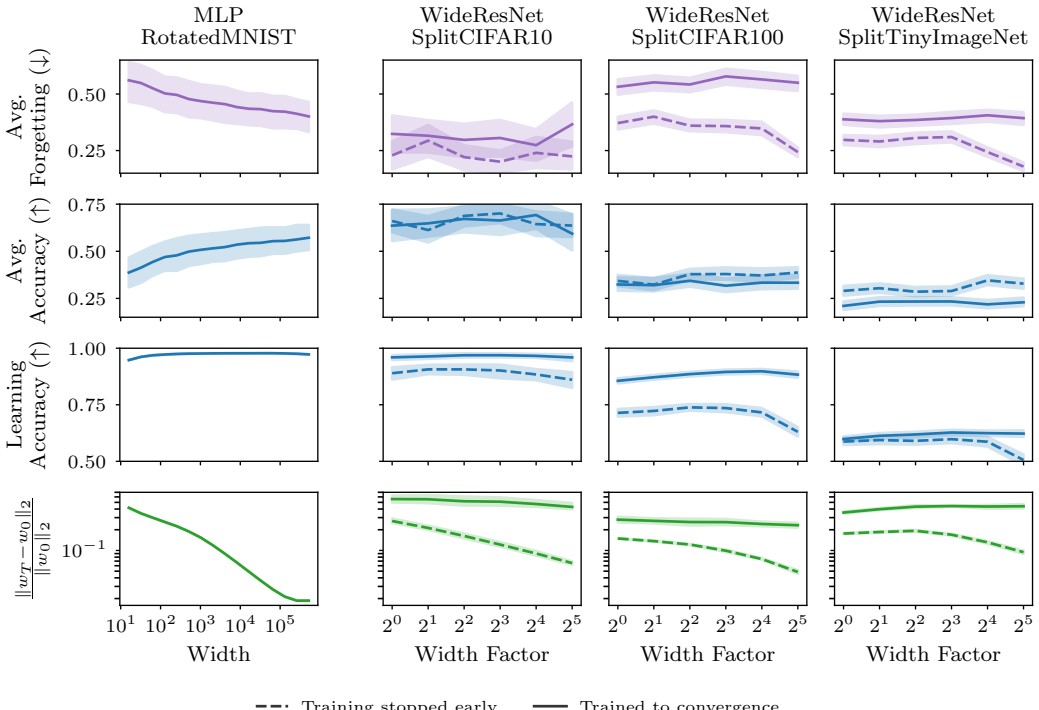

**Figure 7:** *Catastrophic forgetting of wide NNs in theory and practice.* As the width of the shallow MLP approaches the kernel regime, average forgetting decreases, while average accuracy increases. Similar observations hold for WideResNets if trained for a few of epochs only—consistent with experiments by Mirzadeh et al. (2022). However, if they are trained to convergence on each task, resulting in increased learning accuracy, *forgetting does not decrease with width.* This suggests that architectures in practice are not actually wide enough to reduce forgetting via the kernel regime.

## 4 CONCLUSION

In this work, we empirically evaluated whether predictions about the behavior of overparametrized neural networks through the theoretical framework of the neural tangent kernel hold in architectures used in practice. We considered three different areas in which the kernel regime either makes predictions about the behavior of a neural network or informs algorithmic choices. We find that across optimization, uncertainty quantification, and continual learning, theoretical statements in the infinite-width limit do not translate to observable phenomena or improvements in practical architectures with realistic widths. For optimization, we found that such architectures are not sufficiently close to linear to enjoy fast convergence from a second-order method as predicted by existing theory. For uncertainty quantification, we found that controlling the exploration-exploitation trade-off in a sequential decision-making problem via assumptions based on the kernel regime led to performance only marginally better than a random baseline. Finally, in continual learning, we found that wide neural networks as used in practice, if fully trained, do not actually forget less catastrophically.

This observed disconnect between theory and practice leads to two important conclusions. First, our theoretical understanding of the behavior of large-scale overparametrized neural networks is still limited and in particular restricted to architectures that do not resemble those used in practice. This paper is empirical evidence to that effect and thus calls for an effort to improve our understanding by developing a theory under more practically relevant assumptions. Second, algorithms motivated by the neural tangent kernel theory should be scrutinized closely in terms of their practical performance, and researchers should be careful in basing their ideas too strongly on the lazy training regime in the infinite-width limit. We hope in this way our negative results can serve as a cautionary tale and will ultimately benefit *both* the theory and practice of deep neural networks.

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

## APPENDIX A    EXPERIMENT DETAILS

### A.1    TRAINING: SECOND-ORDER OPTIMIZATION

**Conditions for Fast Convergence of NGD**    For (7), we use Jacobian-vector and vector-Jacobian products to compute the Gram matrix's smallest eigenvalue with an iterative sparse eigensolver (Lehoucq et al., 1998). We noticed that such solvers exhibit slow convergence for small eigenvalues out of the box and require additional techniques. As the Gram matrices we investigate here are relatively small, we explicitly computed and decomposed them instead. However, the implicit approach scales beyond that. Likewise, we obtain spectral norms from a partial singular value decomposition which also relies on matrix-free multiplication.

### A.2    UNCERTAINTY QUANTIFICATION: NEURAL BANDITS

We use a two-hidden-layer MLP with width $m = 100$ unless specified otherwise. We use the standard parametrization and initialization in PyTorch—see Section B for a comparison with a different parametrization. To obtain the MAP estimate, we train for 500 epochs using a batch size of 128 and the AdamW optimizer with learning rate 0.001 and weight decay 0.01.

To quantify uncertainty via the empirical NTK, we do a Laplace approximation using the `laplace-torch` library (Daxberger et al., 2021). We use the Kronecker-factored generalized Gauss-Newton to approximate the Hessian. Furthermore, we tune the prior precision via either post-hoc or online marginal likelihood optimization, following Daxberger et al. (2021). We use 10 observations using a random policy as the initial dataset for training the neural network and doing the Laplace approximation. Furthermore, we retrain and perform Bayesian inference every 100 iterations.

We use standard UCI bandits benchmark datasets to compare the algorithms we considered, following (Zhou et al., 2020; Nguyen-Tang et al., 2022; Gupta et al., 2022; Zhang et al., 2021). See Table 1 for details. All experiments were repeated for five random seeds.

**Table 1:** Datasets used in our neural bandit experiment.

|                      | magic | avila | letter | covertype | bean | pendigits |
|----------------------|-------|-------|--------|-----------|------|-----------|
| Input dim. $D$       | 10    | 10    | 16     | 54        | 16   | 16        |
| Num. of classes $C$  | 2     | 12    | 26     | 7         | 7    | 10        |

### A.3    CONTINUAL LEARNING: CATASTROPHIC FORGETTING

The experiment on continual learning discussed in Section 3.3 was carried out using the `Avalanche` library (Lomonaco et al., 2021) on an NVIDIA GeForce RTX 2080 GPU. As models, we chose a 2-layer neural net with ReLU non-linearities (MLP) and a WideResNet (depth $L = 10$) with varying widening factors based on the implementation by Kuen (2017). The benchmark datasets we used are standard benchmarks from continual learning and are described below:

| | |
|---|---|
| *RotatedMNIST*: | Tasks correspond to rotated MNIST digits at varying degrees from 0 to 180 in 22.5-degree increments resulting in nine tasks in total. |
| *SplitCIFAR10*: | Each task corresponds to training on a previously unseen set of 2 out of the total 10 classes of CIFAR10. |
| *SplitCIFAR100*: | Each task corresponds to training on a previously unseen set of 5 out of the total 100 classes of CIFAR100. |
| *SplitTinyImageNet*: | Each task corresponds to training on a previously unseen set of 10 out of the total 200 classes of TinyImageNet. |

The exact training hyperparameters we chose are summarized in Table 2. All experiments were repeated for five different random seeds.

**Table 2:** Training hyperparameters for the continual learning experiment from Section 3.3.

| Model | Depth | Dataset | Tasks | Optim. | Learn. Rate | Mom. | Weight Decay | Batch Size | Epochs |
|---|---|---|---|---|---|---|---|---|---|
| MLP | 2 | RotatedMNIST | 9 | SGD | $10^{-4}$ | 0.9 | $10^{-4}$ | 32 | 5 |
| WideResNet | 10 | SplitCIFAR10 | 5 | SGD | $10^{-1}$ | 0.9 | $10^{-4}$ | 128 | 5/50 |
| WideResNet | 10 | SplitCIFAR100 | 20 | SGD | $10^{-1}$ | 0.9 | $10^{-4}$ | 128 | 5/50 |
| WideResNet | 10 | SplitTinyImageNet | 20 | SGD | $10^{-1}$ | 0.9 | $10^{-4}$ | 128 | 5/50 |

## APPENDIX B    ADDITIONAL EXPERIMENTAL RESULTS

### B.1    UNCERTAINTY QUANTIFICATION: NEURAL BANDITS

In the neural bandit experiment we used the standard parametrization (SP)—the default in PyTorch, also known as the NNGP parametrization (Lee et al., 2018). We provide an additional result comparing NeuralUCB with the SP and the neural tangent parametrization (NTP) in Fig. 8.[9] We observe that they give very similar results. In conjunction with the fact that the SP is the *de facto* standard in practice—i.e. it is the default in PyTorch—these facts justify our choice of parametrization in the bandit experiment in Section 3.2.

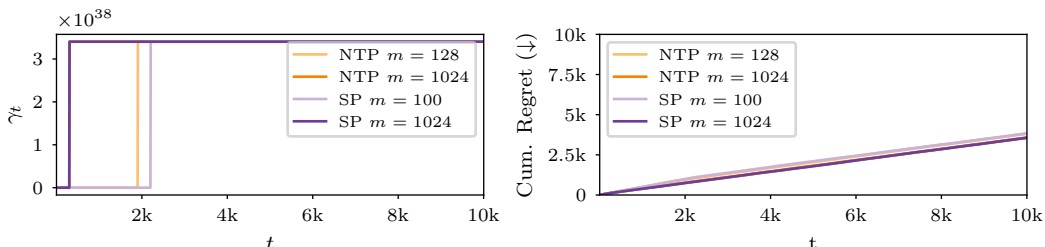

**Figure 8:** NeuralUCB with the standard parametrization (SP) and the neural tangent parametrization (NTP) on the `magic` dataset. Overexploration can be seen in both parametrizations, resulting in a similarly poor performance.

### B.2    CONTINUAL LEARNING: CATASTROPHIC FORGETTING

We provide the detailed results from our continual learning experiment in Figs. 9 and 10 and Fig. 11. In the toy setting of an MLP trained on RotatedMNIST in Fig. 9, where widths are orders of magnitude larger than in practice, the amount of forgetting decreases with width for each task. This is in line with the hypotheses for why in the kernel regime catastrophic forgetting should be mitigated, namely increasingly orthogonal gradients across tasks and minimal changes in the weights of the network.

In the practical setting for WideResNets trained on SplitCIFAR100 and SplitTinyImageNet, we see a similar, albeit less pronounced, phenomenon for networks only trained for a few (here five) epochs. However, peak accuracy per task drops significantly–more so for wider networks. This indicates they are not trained sufficiently. The amount of forgetting in this "short training" setting decreases primarily due to a drop in peak accuracy, and less of a decrease in performance as the NN is trained on new tasks. However, if WideResNets of increasing width are trained to convergence (here fifty epochs), i.e. to high learning accuracy, a decrease in forgetting with width is no longer observable. In fact, the higher peak accuracy leads to larger forgetting, because the difference between peak and final accuracy is increased.

---

[9]The term "parametrization" here is rather misleading (Kristiadi et al., 2023a), but we follow the standard naming convention for clarity.

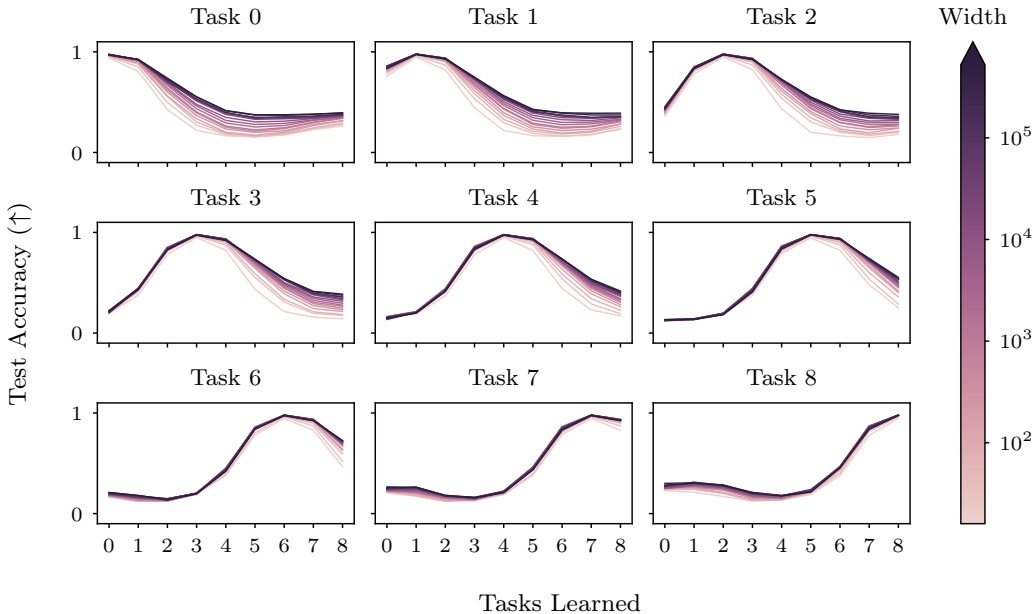

**Figure 9:** Accuracy of a fully-connected NN with ReLU non-linearities, depth $L = 2$ and increasing width trained sequentially on different tasks from the RotatedMNIST dataset.

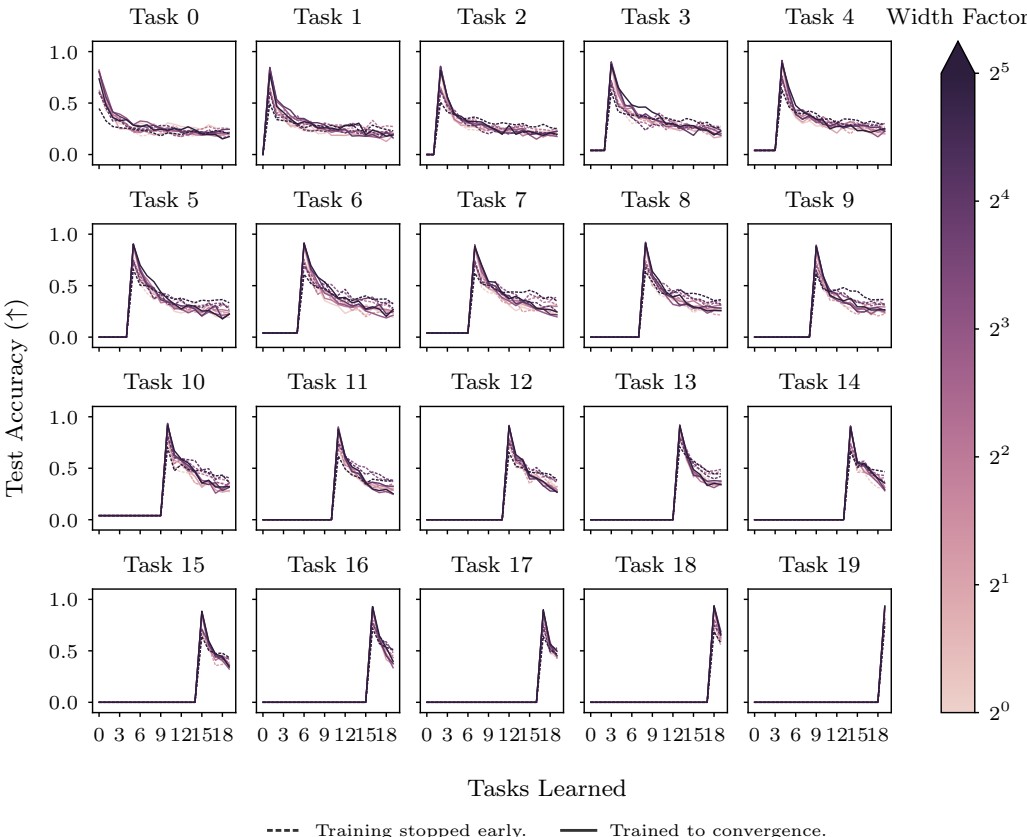

**Figure 10:** Accuracy of a WideResNet with depth $L = 10$ and increasing width factor trained sequentially on different tasks from the SplitCIFAR-100 dataset.

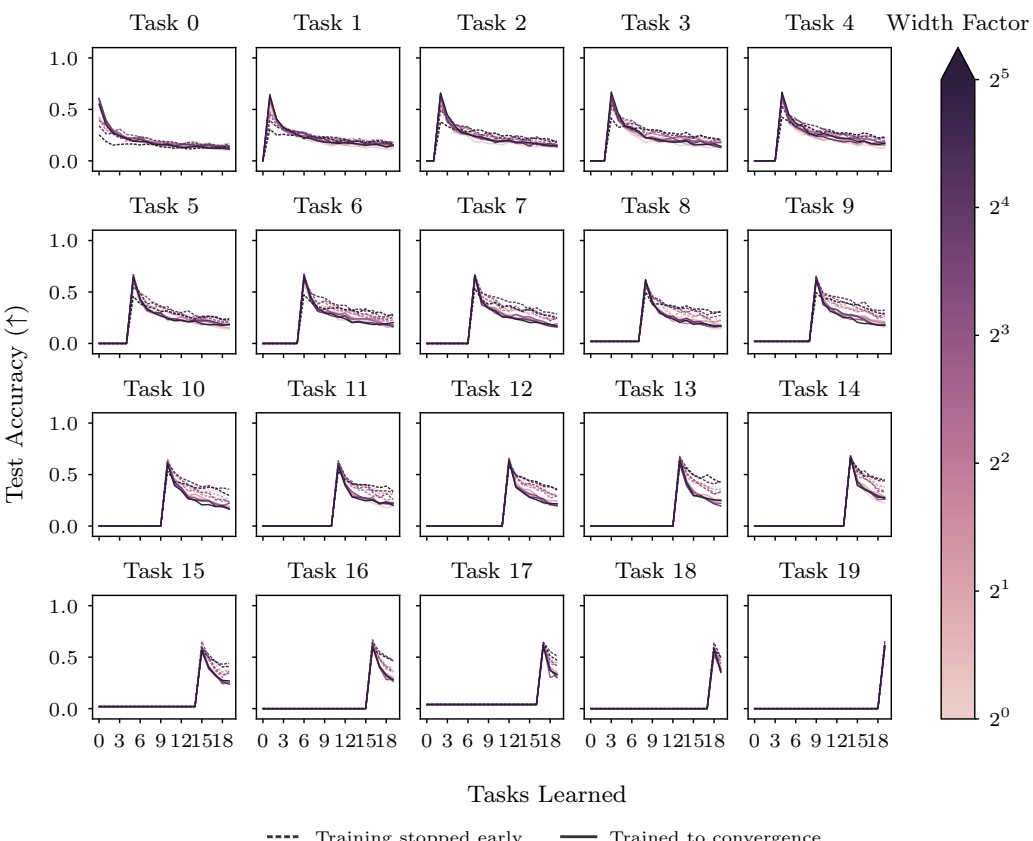

**Figure 11:** Accuracy of a WideResNet with depth $L = 10$ and increasing width factor trained sequentially on different tasks from the SplitTinyImageNet dataset.

