# OpenReview forum: "On the Disconnect Between Theory and Practice of Overparametrized Neural Networks"
_ICLR.cc/2024/Conference — Submitted to ICLR 2024_

### Official Review · Reviewer_cgm5 · 2023-10-25

**Soundness:** 3 good
**Presentation:** 3 good
**Contribution:** 2 fair
**Rating:** 5
**Confidence:** 3

**Summary:**

This paper empirically verifies the linearization effect of wide neural networks and the challenges of applying it to practical neural networks in downstream applications. Specifically, the authors demonstrate the gap between theory and practice across three domains: Optimization (Section 3.1), Uncertainty Quantification (Section 3.2), and Continual Learning (Section 3.3). One of the fundamental reasons for this phenomenon is that practical neural networks do not satisfactorily meet the Stable Jacobian condition (Equation 8). Based on this issue, the authors highlight potential pitfalls that can arise from ideas rooted in linearization for each task.

**Strengths:**

* The paper is easy-to-follow, and its intended claims are clear.
* The empirical gap between linearized NNs and practical NNs tackled in this paper is timely. Additionally, the authors' approach to the problem (Stability of Jacobian) is intriguing.
* The claims made by the authors are adequately supported through experiments.
* The authors have clearly delineated the potential and limitations of their claims.

**Weaknesses:**

* While each of the authors' claims is clear, the connections between them are somewhat disjointed. Specifically, it is challenging to perceive Section 3.1-3.2's issues as problems since, in reality, wide NNs experience less catastrophic forgetting, as suggested in Section 3.3. Therefore, combining Sections 3.1 and 3.2 into one section and separating Section 3.3 might help avoid this confusion.

* The title of Section 3.1, "Training: Second-order Optimization", doesn't accurately convey the content discussed within. The empirical evidence in Section 3.1 (Figures 2-3) primarily supports the instability of the Jacobian in finite NNs. While this might be one of the reasons second-order optimization methods fail when applied to NNs, the paper doesn't seem to present direct experiments that validate this. Moreover, second-order optimization can fail for more than one reason: Slow convergence and Poor generalization. The authors do not specify which of these reasons is attributed to the unstable Jacobian. To avoid confusion, it's recommended that the title of Section 3.1 be revised.

**Questions:**

* Have you considered restructuring the sections to more coherently present the content, possibly merging Sections 3.1 and 3.2, and separating Section 3.3?
* Second-order optimization can exhibit issues like slow convergence and poor generalization. Could you specify which of these issues the unstable Jacobian directly contributes to, based on your research?

If the issues in Weakness & Questions are addressed appropriately, I will raise the score.

---

> ### Author Response · Authors · 2023-11-17
> **Author Response**
>
> **Thank you for your review and feedback!** If we could answer your questions, we would appreciate if you'd consider updating your score. If anything remains unclear, please let us know.
>
> ## Detailed Response
>
> > While each of the authors' claims is clear, the connections between them are somewhat disjointed. Specifically, it is challenging to perceive Section 3.1-3.2's issues as problems since, in reality, wide NNs experience less catastrophic forgetting, as suggested in Section 3.3. Therefore, combining Sections 3.1 and 3.2 into one section and separating Section 3.3 might help avoid this confusion.
>
> Could you clarify what you mean by "[...] it is challenging to perceive Section 3.1-3.2's issues as problems since, in reality, wide NNs experience less catastrophic forgetting, as suggested in Section 3.3."?
> While the issues described in Sections 3.1 and 3.2 have the same underlying assumptions (that the networks behave according to the kernel regime), they all consider different domains (training, uncertainty quantification and continual learning). Only Section 3.3 concerns itself with catastrophic forgetting.
>
> All three sections (training, uncertainty quantification, and continual learning) make assumptions based on the kernel regime. For provably faster optimization via natural gradient descent, we find that practical architectures are not wide enough to be close to the kernel regime and thus satisfy Jacobian stability. In the neural bandit setting, we find that sequential actions taken based on the uncertainty as given by the empirical NTK perform no better than random. In theory if the network was sufficiently close to the kernel regime, it should achieve near-optimal regret (Zhou et al., 2020). Finally, in the continual learning setting catastrophic forgetting can be predicted and mitigated if the network is near the kernel regime. In practice, at first glance wider networks seem to mitigate forgetting (as observed by Mirzadeh et al., 2022), but based on our experiments only if the networks are not trained to high accuracy, which is clearly desirable in practice. Therefore all of these sections show that drawing conclusions from the kernel regime about the behavior of neural networks used in practice can be problematic. This is the observed disconnect we describe.
>
> > Have you considered restructuring the sections to more coherently present the content, possibly merging Sections 3.1 and 3.2, and separating Section 3.3?
>
> We believe all three sections 3.1-3.3 fit semantically under one heading, since they all assume the networks behave according to the kernel regime, while in practice with realistic widths they behave very differently. For the final version we will improve the introduction to Section 3 to make that very explicit. Would a different title for Section 3 rather than "Connecting Theory and Practice" help make it more clear what we are describing in Section 3?
>
> > The title of Section 3.1, "Training: Second-order Optimization", doesn't accurately convey the content discussed within. The empirical evidence in Section 3.1 (Figures 2-3) primarily supports the instability of the Jacobian in finite NNs. While this might be one of the reasons second-order optimization methods fail when applied to NNs, the paper doesn't seem to present direct experiments that validate this. Moreover, second-order optimization can fail for more than one reason: Slow convergence and Poor generalization. The authors do not specify which of these reasons is attributed to the unstable Jacobian. To avoid confusion, it's recommended that the title of Section 3.1 be revised.
> >
> > Q: Second-order optimization can exhibit issues like slow convergence and poor generalization. Could you specify which of these issues the unstable Jacobian directly contributes to, based on your research?
>
> You are completely correct that second-order methods may not only converge slowly, but also demonstrate poor generalization or other issues.
> We only consider the _speed of convergence_ of the training loss in our work. As we describe on page 4 in the paragraph "Fast Convergence of NGD in the Kernel Regime", a network in the kernel regime, if trained with natural gradient descent (an approximate second-order method), provably exhibits fast convergence (Theorem 3, Zhang et al. 2019).
> The proof by Zhang et al. (2019) assumes conditions (7) and (8), whereby (8), the Jacobian stability, is the condition which is satisfied if the network is sufficiently close to the kernel regime. We experimentally demonstrate that this assumption is not satisfied for practical architectures (see Figure 2b).
> While second-order methods may perform well for deep learning, our findings show this cannot be explained via the kernel regime. To reflect better what we describe in Section 3.1, we changed the title to "Training: Conditions for Fast Convergence of Second-Order Optimization".

---

> > ### Comment · Reviewer_cgm5 · 2023-11-22
> >
> > *  In practice, at first glance wider networks seem to mitigate forgetting (as observed by Mirzadeh et al., 2022), but based on our experiments only if the networks are not trained to high accuracy, which is clearly desirable in practice. Therefore all of these sections show that drawing conclusions from the kernel regime about the behavior of neural networks used in practice can be problematic. This is the observed disconnect we describe.
> >
> > * I wanted to point out that while Sections 3.1 and 3.2 seem homogeneous as they use the same underlying assumptions, Section 3.2 appears heterogeneous due to the absence of such a discussion. According to Section 3.3, for sufficiently trained NNs, width does not help. However, this is only a counterexample to the conclusions in the kernel regime literatures and does not provide specific insights. For instance, why does width not help only in well-trained NNs? Why do less-trained NNs benefit from width? Which parts of the claim by Mirzadeh et al., 2022, are incorrect? For these reasons, the conclusions of Section 3.3 seem to be just an interesting counterexample to the theme of Connecting Theory and Practice in Section 3, without including sufficient discussion on the causes.
> >
> > *  While second-order methods may perform well for deep learning, our findings show this cannot be explained via the kernel regime. To reflect better what we describe in Section 3.1, we changed the title to "Training: Conditions for Fast Convergence of Second-Order Optimization".
> >
> > * Similar to Section 3.3, Section 3.2 only shows that the success of second-order optimization methods cannot be explained through the kernel regime. However, real-world second-order optimization methods [1,2] often demonstrate faster convergence than first-order optimization methods, and they do not base their theoretical superiority on the kernel regime. Considering this, the experiments and conclusions of Section 3.2 may not feel significant.
> >
> > Considering the above points, I believe this paper deserves a rating higher than 3 but not significant enough for a 6. I will maintain my current score.
> >
> > [1] Martens, James, and Roger Grosse. "Optimizing neural networks with kronecker-factored approximate curvature." International conference on machine learning. PMLR, 2015.
> > [2] Liu, Hong, et al. "Sophia: A Scalable Stochastic Second-order Optimizer for Language Model Pre-training." arXiv preprint arXiv:2305.14342 (2023).
> > [3] Roulet, Vincent, and Mathieu Blondel. "Dual Gauss-Newton Directions for Deep Learning." arXiv preprint arXiv:2308.08886 (2023).

---

> > > ### Author Response · Authors · 2023-11-22
> > > **Response to Follow-up**
> > >
> > > Thanks a lot for your reply! We're confident that the issues you raised can easily be solved.
> > >
> > > First of all, even though Sec. 3.1-3.3 fall into the same theme in that they based their assumptions in the kernel regime, we take your point that they can potentially be clearer as separate sections.
> > > Our main goal is indeed to show independent examples of where the kernel-regime assumption fails when used to explain practical phenomena (Sec. 3.1, 3.3) or to develop a practical method (Sec. 3.2).
> > > We are happy to separate them into three sections as you suggested.
> > >
> > > As to your new specific questions:
> > >
> > > **Continual Learning**
> > >
> > > > For instance, why does width not help only in well-trained NNs?
> > >
> > > Large-scale networks used in practice are simply not wide enough to operate in the lazy regime. To see this, consider the bottom row of Figure 7, which shows the relative change in parameters. Well-trained NNs have significant relative change in their parameters during training even for large (realistic) widths. For them to benefit from decreased forgetting via the kernel regime, widths would presumably have to be orders of magnitude larger (which is prohibitive in practice).
> > >
> > > As we write in Section 3.3., paragraph Experimental Results: "[...] training networks to high accuracy increases average forgetting since the peak performance across tasks increases. This can be explained by the fact that they are not actually operating in the kernel regime. The relative change of the weights during training remains large even as width increases beyond what is used in practice, meaning the networks are still adapting to unseen tasks."
> > >
> > > > Why do less-trained NNs benefit from width?
> > >
> > > As we write in Section 3.3., paragraph Experimental Results: "NNs that are trained insufficiently show a decrease in forgetting as they become wider. But, this decrease is primarily due to lower learning accuracy, and thus a smaller gap between peak accuracy and minimum accuracy across tasks (see Section B.2)."
> > > The interpretation is: Forgetting is reduced since peak performance is reduced when training less, not because the network is in the lazy regime.
> > >
> > > > Which parts of the claim by Mirzadeh et al., 2022, are incorrect?
> > >
> > > The overall claim that catastrophic forgetting can be reduced with width is correct. However, according to our experiments the claim that this applies to _architectures used in practice_ (e.g. WideResNet) is incorrect. We could only reduce forgetting by training only briefly and thus sacrificing performance. Forgetting was _not_ reduced with width if the networks were trained fully. This was not necessary in the toy example (first column of Fig 7), where we could scale performance with width and reduce forgetting via the kernel regime.
> > >
> > > **Second-order Optimization**
> > >
> > > > Similar to Section 3.3, Section 3.2 only shows that the success of second-order optimization methods cannot be explained through the kernel regime. However, real-world second-order optimization methods [1,2] often demonstrate faster convergence than first-order optimization methods, and they do not base their theoretical superiority on the kernel regime.
> > >
> > > You are correct that we show that the performance of second-order methods cannot be explained through the kernel regime.
> > > While optimizers like Sophia don't directly assume the kernel regime for their theoretical analysis, they still assume strict convexity and make assumptions on the curvature of the loss landscape (e.g. Assumptions 4.1 and 4.2 in [2] are satisfied in the kernel regime in the case of regression). Works like [3], that seek to understand why approximate NGD methods such as KFAC [1] might work well in practice, link their results to the convergence behavior of NGD in the NTK regime. Again, we believe that our findings imply that such an analysis may not explain the benefits of such methods that have been reported and that we need new perspectives to explain their properties.
> > >
> > > ---
> > >
> > > We will state these arguments more clearly in the final version of the paper.
> > >
> > > We hope you will consider our response in your final assessment of our work.
> > >
> > > ---
> > >
> > > **References:**
> > >
> > > - [1] Martens, J., & Grosse, R. (ICML, 2015). Optimizing neural networks with Kronecker-factored approximate curvature.
> > > - [2] Liu, H., Li, Z., Hall, D., Liang, P., & Ma, T. (arXiv, 2023). Sophia: a scalable stochastic second-order optimizer for language model pre-training.
> > > - [3] Karakida, R., & Osawa, K. (NeurIPS, 2020). Understanding approximate fisher information for fast convergence of natural gradient descent in wide neural networks.

---

### Official Review · Reviewer_PQv1 · 2023-10-29

**Soundness:** 2 fair
**Presentation:** 3 good
**Contribution:** 2 fair
**Rating:** 5
**Confidence:** 2

**Summary:**

The paper is experimental. It is dedicated to testing assumptions of several previous papers (mostly theoretical). These previous papers exploit the NTK theory and make a number of predictions about the behavior of NNs. The main claim of the paper is that these predictions do not hold in a practical setting. Three examples of such predictions are considered: 1) the claim that second-order optimization algorithm has some advantages over 1st order in the infinite width limit (the claim from Zhang et al. (2019)); 2) setting the exploration parameter in Neural Uncertainty Quantification according to a formula by Zhou et al. (2020), that was inspired by the NTK theory, 3) the claim that "increasing
the width of a neural network reduces catastrophic forgetting".

**Strengths:**

The setup of experiments is clear, their description is complete. Experimental results are convincing.

**Weaknesses:**

I doubt the main claim of the paper: "observed disconnect between theory and practice calls into question the practical relevance of the infinite-width limit". It seems to me that this claim does not follow logically from the reported experiments.

The experimental results of Section 3.1 seem to refute the theory of Zhang et al. (2019), rather than the NTK theory. Experimental evidence that Stable Jacobian conditions (7) and (8) are never satisfied does not "call into question the practical relevance of the infinite-width limit". It refutes the claim that NGD is better than GD, but not the NTK theory. Logically, the NTK theory does not claim that (7) and (8) should be satisfied.

The same holds for the Neural Contextual Bandits experiments. It seems that Section 3.2 shows that the formula for the exploration parameter from Zhou et al. is not very good in a practical setting (probably, because the assumptions of Zhou et al are not satisfied). But this does not mean that "the practical relevance of the infinite-width limit" is in question.

Concerning catastrophic forgetting, there is no discussion of why the previous research (e.g. Mirzadeh et al. (2022)) somehow contradicts to experimental results of the paper.

**Questions:**

The general question is: how the irrelevance of the NTK theory follows from the fact that certain assumptions of a theory developed in a previous paper (e.g. Stable Jacobian conditions (7) and (8)) are not satisfied in current experiments.

---

> ### Author Response · Authors · 2023-11-17
> **Author Response**
>
> **Thank you for your taking the time to review our paper.** If we could answer your questions to your satisfaction, we would appreciate if you would consider updating your score.
>
> ## Detailed Response
>
> > The experimental results of Section 3.1 seem to refute the theory of Zhang et al. (2019), rather than the NTK theory.
>
> We do not refute the theory of Zhang et al. (2019). Their theory is correct, but the required width to reach the kernel regime, such that conditions (7) and (8) are satisfied are out of reach in practical architectures.
>
> > Logically, the NTK theory does not claim that (7) and (8) should be satisfied.
>
> This is incorrect. In the limit of large-width, Jacobian stability (8) is satisfied. In fact, as Zhang et al. (2019) write below "Condition 2" in their paper, eqn (8) with small $C$ implies that the NN is close to a linearized NN at initialization. The NTK is simply the function-space view of this property. In fact, Lee et al. (2019) use Jacobian stability (formulated as a Lipschitz smoothness condition, see paragraph below Condition 2 in Zhang et al. (2019)) to prove the convergence of the empirical NTK to the empirical NTK at initialization at the rate we give in eqn (6) (c.f. eqn S50 and eqns S69-S72). Condition 1 (eqn 7) is simply a regularity condition, which is violated e.g. if there are duplicate datapoints in the dataset.
>
> To summarize, Jacobian stability is simply a "weight-space view" condition for when a NN is in the kernel regime. In Figures 2 and 3 we show that both for wide toy architectures and for WideResNets with widths as used in practice, Jacobian stability is _not_ satisfied. Therefore, conclusions drawn from the kernel regime do not apply, in particular, NGD does not necessarily converge faster than gradient descent.
>
> > The general question is: how the irrelevance of the NTK theory follows from the fact that certain assumptions of a theory developed in a previous paper (e.g. Stable Jacobian conditions (7) and (8)) are not satisfied in current experiments.
>
> Our claim is not that NTK theory is irrelevant, rather that its assumptions are not satisfied in practice. Therefore making assumptions based on the kernel regime can be misleading and to suboptimal algorithms. We demonstrate this across three different domains, pointing out a disconnect between theory and practice. Specifically for Section 3.1: If practical architectures were well-described by NTK theory, Jacobian stability (8) would be satisfied, which is not what we see in our experiments.
>
> > The same holds for the Neural Contextual Bandits experiments. It seems that Section 3.2 shows that the formula for the exploration parameter from Zhou et al. is not very good in a practical setting [...]. But this does not mean that "the practical relevance of the infinite-width limit" is in question.
>
> As you correctly point out, we observe experimentally that the exploration parameter choice of Zhou et al. (2020) does not lead to competitive results. However, they prove that under the assumption of sufficiently large width (s.t. the NN is in the kernel regime), their choice of $\gamma_t$ leads to near-optimal regret $\tilde{O}(\sqrt{T})$ with high prob. (see Thm 4.5, Zhou et al. 2020). Now, to explain our experimental results, either their theorem is incorrect, which we do not claim, or the NN does _not_ in fact behave as if in the kernel regime, which is our contention. Therefore the infinite-width limit is not informative for this case.
>
> In the sequential decision making setting, where UQ is crucial, making assumptions derived from the kernel regime can be counterproductive. Instead, we propose to leverage UQ tools that are designed for finite-width neural NNs as deployed in practice, such as the Laplace approximation. We emphasize that this does not mean that the NTK theory is not important for understanding NNs.
> We only claim that it is more effective to use finite-width approximations/theories for finite-width problems.
>
> > Concerning catastrophic forgetting, there is no discussion of why the previous research (e.g. Mirzadeh et al. (2022)) somehow contradicts the experimental results of the paper.
>
> As we write in Section 3.3 (paragraph "Experiment Results" and caption of Figure 7) the results for the shallow MLP seem to confirm that catastrophic forgetting decreases with width as Mirzadeh et al. (2022) suggest. However, they also found the same to hold for WideResNet on SplitCifar100. A finding which based on our experimental results was due to early stopping, _not_ the lazy regime. Therefore, if practical architectures are trained to high accuracy, large width does not mitigate forgetting (see Figure 7). Therefore, as we write in the caption of Figure 7: "[...] architectures in practice are not actually wide enough to reduce forgetting via the kernel regime." Could you clarify in what way you believe the differences in experimental results from Mirzadeh et al. (2022) and our paper were not discussed?

---

### Official Review · Reviewer_59VJ · 2023-10-31

**Soundness:** 4 excellent
**Presentation:** 4 excellent
**Contribution:** 3 good
**Rating:** 8
**Confidence:** 4

**Summary:**

This paper empirically investigates whether the theoretical predictions and assumptions based on infinite width NTK theory holds for practical widths encountered in common architectures on relevant tasks. In particular, the authors consider three areas - optimization, uncertainty quantification, and continual learning.

* For optimization, second-order methods like natural gradient are hypothesized to have faster convergence compared to first-order methods like SGD when networks approach the kernel regime. However, the authors find that common architectures do not satisfy the necessary conditions (such as Jacobian stability) for this to hold.
* In uncertainty quantification, controlling the exploration-exploitation tradeoff in neural bandits via the NTK theory leads to poor performance, while other methods like online evidence maximization work better.
* In continual learning, wide networks appear to forget less catastrophically only if not trained to high accuracy per task. So increasing width does not mitigate forgetting for practical architectures.

**Strengths:**

The paper is very well written and easy to follow, even if the reader does not have a very strong background in kernel and gaussian process theory. The question it addresses in an important one. Although prior papers (Fort et al arxiv.org/abs/2010.15110, Atanasov et al arxiv.org/abs/2212.12147) have studied the distinctions between infinite and finite width networks, this paper contributes novel insights. I do however recommend that the authors do a more thorough literature review and cite prior papers studying infinite vs finite width networks, and cite the above mentioned papers. The reviewer especially appreciates the empirical tests in the neural bandit and catastrophic forgetting settings, which are not commonly encountered by theorists working on NTK-related analysis.

**Weaknesses:**

My primary critique is the lack of distinction made between width and feature learning strength. Although it is true that in standard/NTK parameterization, wide networks converge to the NTK (as in Jacot et al), alternative parameterizations have since been considered. Firstly, (Chizat et al http://arxiv.org/abs/1812.07956) have shown even finite width networks can be reparameterized to make them behave as kernels and have identified a "laziness" parameter $\alpha$ that can control feature learning strength at any width.

Most importantly, the mean field parameterization (Mei & Montanari https://arxiv.org/abs/1804.06561, several other concurrent works) also known as $\mu$-parameterization (Yang and Hu https://arxiv.org/abs/2011.14522) allows networks to learn features at infinite width. Moreover, (Vyas et al http://arxiv.org/abs/2305.18411) have shown that finite width networks approach the infinite-width feature learning limit very quickly and efficiently. This would imply that in that parameterization such a distinction between wider and narrower networks would be substantially less prominent. One consequence of this is hyperparameter transfer across widths (Yang and Hu http://arxiv.org/abs/2203.03466 ).

I do not expect the authors to redo experiments in this alternative parameterization (though that would certainly be an interesting follow up). It would be very good, however, to distinguish between *overparameterized theory* not being representative of realistic finite-width networks (which I think is an incorrect claim) vs *lazy training at large widths* being non-representative of realistic finite-width networks (which is the claim that the paper very nicely supports). A few sentences in the introduction making this explicit would be very welcome.

**Questions:**

For optimization, besides stability conditions, were there other signs like faster convergence that second-order methods may work better? Or was SGD consistently better?

What happens if you make the tasks more similar in the continual learning setting? Does the NTK theory begin to hold?

---

> ### Author Response · Authors · 2023-11-17
> **Author Response**
>
> **Thank you for your positive feedback!** We will make it more explicit in the final version of the paper that we consider the _lazy_ training regime.
> We hope we could answer any questions you raised below. If anything should remain unclear, we would be glad to address it.
>
> ## Detailed Response
>
> > My primary critique is the lack of distinction made between width and feature learning strength. [...] I do not expect the authors to redo experiments in this alternative parameterization (though that would certainly be an interesting follow up). It would be very good, however, to distinguish between overparameterized theory not being representative of realistic finite-width networks (which I think is an incorrect claim) vs lazy training at large widths being non-representative of realistic finite-width networks (which is the claim that the paper very nicely supports). A few sentences in the introduction making this explicit would be very welcome.
>
> We will make this distinction more clear both in the introduction and by adding a paragraph to the supplementary (due to space constraints) that explicitly mentions and cites different parametrizations that have been proposed based on the references you provide.
>
> > For optimization, besides stability conditions, were there other signs like faster convergence that second-order methods may work better? Or was SGD consistently better?
>
> To the best of our knowledge there's no empirical comparison of first and second-order methods on large-scale benchmark datasets that shows a clear performance difference.
> Here, we consider theoretical conditions under which second-order methods (specifically NGD) converge _provably faster_ than first-order methods (Zhang et al. 2019; Thm. 1 and paragraph below). These conditions are satisfied in the kernel regime and we test them empirically for architectures used in practice with realistic widths. To test these conditiones it is not necessary to train the network, but only to evaluate the difference in Jacobians at initialization and the Jacobians at parameters in a ball around the initial parameters.
> This is what we test and show in Figure 2 and 3.
>
> > What happens if you make the tasks more similar in the continual learning setting? Does the NTK theory begin to hold?
>
> For toy architectures with very large widths the predictions from NTK theory hold. The parameters change only very little from initialization (and therefore also the learned representation) and catastrophic forgetting is reduced (see Figure 7, left column). Now as you rightly point out, following this reasoning suggests that if tasks are more similar, then catastrophic forgetting should also be reduced, possibly also at much smaller widths. One can see this in Figure 9, for the toy architecture on rotated MNIST (where tasks differ by a rotation of 22.5 degrees). Forgetting is reduced on tasks that are within a few 22.5 degree rotations of each other. It is still larger for smaller widths, but not nearly as large for more similar tasks than for very different tasks (c.f. Task 0: Tasks Learned = (0,1) vs Tasks Learned = (0, 4)).

---

> ### Comment · Reviewer_59VJ · 2023-12-04
> **Reviewer Response**
>
> I thank the authors for their response and continue to support acceptance.

---

### Official Review · Reviewer_9JCy · 2023-10-31

**Soundness:** 3 good
**Presentation:** 3 good
**Contribution:** 3 good
**Rating:** 6
**Confidence:** 3

**Summary:**

The paper tests the practical validity of the theoretical connection between overparameterized (infinitely-wide) neural networks and the NTK regime. It is shown empirically that in many setups used in practice, this connection does not hold for practical network widths.
In turn, this has interesting consequences in several important fields (faster optimization, reliable uncertainty quantification and continual learning).

**Strengths:**

- Important research question and implications. Relevant to many recent studies.
- Well written. Clear.
- Interesting finding in CL showing that early stopping might significantly affect the conclusions drawn from CL experiments where the width is increased.

**Weaknesses:**

1. **Theoretical applicability.** The theory of NTKs requires not only an infinite width, but also a small step size and a large initialization scale (see the original paper by Jacot and "On Lazy Training in Differentiable Programming").
The paper submitted here only considers the effect of increasing the width.
1. **Novelty**: It was hard for me to understand how novel the submitted paper is.
I would appreciate a comparison to "Empirical Limitations of the NTK for Understanding Scaling Laws in Deep Learning".


Moreover, some points were not completely clear to me. See the questions in the following section.

  ---

### Minor remarks (I don't expect any response for the rebuttal on these issues)
- It seems the authors are unaware of [Goldfarb and Hand, AISTATS 2023] that proved theoretically that forgetting decreases with overparameterization for linear models under a random data model. I advise citing that paper.
- In Figure 1(c), perhaps a log-scale is more suitable for the y-axis as well. Also, consider drawing $\frac{1}{\sqrt{m}}$.
- In Figure 2, I suggest writing the depth (i.e., $L=10$) in the legend for the real world architecture as well.
- In Figure 4, consider using a log-scale for the x-axis.
- On page 8 (experimental setup), the authors say they use WideResNets but don't specify the number of layers).

**Questions:**

1. In Section 3.1, can the authors explain in what sense does NGD have "favorable convergence over GD" in theory? Does the paper refer to an appropriate source for this statement? Is this ``strict'' in some way? (otherwise it's unclear why testing only NGD is sufficient to draw conclusions on GD).
1. In Section 3.1 (Page 6), why do the authors use only a subset of the data for the CNN regression experiment?
   (this should be explained in the paper as well).
1. Still in Section 3.1, does it make sense to somehow plot in Figure 2 (e.g., with a horizontal line) the $\lambda_{\min}$ and $C'$ in the limit where the width$\to\infty$ (i.e., in the NTK regime)? Is there perhaps a way to compute that theoretically rather than numerically? I believe it may complete the picture for this experiment.

**Details Of Ethics Concerns:**

None.

---

> ### Author Response · Authors · 2023-11-17
> **Author Response**
>
> **Thank you for your review and detailed feedback!** We included your suggestions in the revised version. We hope we could answer the questions you raised in your review below, if not we are happy to clarify further.
>
> ## Detailed Response
> > It was hard for me to understand how novel the submitted paper is. I would appreciate a comparison to "Empirical Limitations of the NTK for Understanding Scaling Laws in Deep Learning".
>
> While Vyas et al. (2023) briefly discuss generalization as a function of width, they primarily study the NTK in the context of scaling laws, i.e. how quickly generalization error decreases with more training data. In contrast, our paper primarily studies predicted phenomena from the kernel regime as a function of width. Our work focuses on domains where the kernel regime promises to inform better algorithms in practice. More concretely, how to train them efficiently, how to obtain reliable and cheap uncertainty quantification and how to deploy them in a continual learning setting. In that sense our paper complements the findings by Vyas et al. (2023). Not only is the NTK limit not predictive of finite-width NN generalization, but predictions from the kernel regime should also not be used without scrutiny as the basis for better algorithms (training, uncertainty quantification) or guiding principles in practice (continual learning). We've added a citation to Vyas et al. (2023) to the revised version of the paper.
>
> > In Section 3.1, can the authors explain in what sense does NGD have "favorable convergence over GD" in theory? Does the paper refer to an appropriate source for this statement? Is this ``strict'' in some way? (otherwise it's unclear why testing only NGD is sufficient to draw conclusions on GD).
>
> We apologize for not clearly citing this statement, which is detailed in [1] ([paragraph below Equation (11)](https://arxiv.org/pdf/1905.10961.pdf), $n$ is the number of data points and $\lambda_0$ is defined in footnote 6 of our paper):
>
> > Compared to analogous bounds for gradient descent [Du et al., 2018a, Oymak and Soltanolkotabi, 2019, Wu et al., 2019], we improve the maximum allowable learning rate from $\mathcal{O}(1/n)$ to $\mathcal{O}(1)$ and also get rid of the dependency on $\lambda_0$. Overall, NGD (Theorem 3) gives an $\mathcal{O}(\lambda_0 / n)$ improvement over gradient descent.
>
> We will make this more clear in the final version of the paper.
>
> > In Section 3.1 (Page 6), why do the authors use only a subset of the data for the CNN regression experiment? (this should be explained in the paper as well).
>
> We use only a subset of the data for the regression experiment in Section 3.1. due to computational constraints at large widths. For example, for the WRN computing one data point for large widths and $N=2000$ requires roughly 3 GPU days.
>
> More specifically, the JVPs and VJPs required to multiply with the NTK matrix require looping over the data in batches, performing forward and backward passes. For NNs with very large widths those operations quickly become expensive (note that for fully-connected NNs the number of parameters scale as $O(P)=O(m^L)$ with width, and JVP and VJP costs scale with number of parameters). Since we want to explore large widths, our only choice to reduce cost is to use less data. To make sure that this does not corrupt our experiments, we compared the conditions for two different sub-set sizes in Fig. 3 which confirms that using more data leads to weaker linearity (as predicted by the theory too).
>
> > Still in Section 3.1, does it make sense to somehow plot in Figure 2 (e.g., with a horizontal line) the $\lambda_{\text{min}}$ and $C'$ in the limit where the width $\to \infty$ (i.e., in the NTK regime)? Is there perhaps a way to compute that theoretically rather than numerically? I believe it may complete the picture for this experiment.
>
> We can compute $\lambda_{\text{min}}$ theoretically for the toy architectures, since the limiting kernel is known.
> The Gram matrix converges to $\mathbf{G}^\infty_{ij} = \mathbf{x}_i^T \mathbf{x}_j\frac{\pi - \arccos(\mathbf{x}_i^T \mathbf{x}_j)}{2\pi}$ (see Zhang et al. (2019), Definition 1). We've computed the minimum eigenvalue of this matrix and added it as a dashed horizontal line to the plot (see Figure 2a in the revised paper). For Figure 2b, we can draw $C'$ in the limit of infinite width, since as we write in the paper below eqn. 8, $C \to 0$. We've drawn $C'=0.5$ as a black dashed line in Figure 2 to indicate the threshold for which the Jacobian is sufficiently stable for rapid convergence of NGD.

---

> > ### Comment · Reviewer_9JCy · 2023-11-22
> >
> > I thank the authors for their detailed response.
> >
> > I have gone through the discussion thus far.
> > Although I have no further questions, I would like to discuss with AC and other Reviewers before finalizing my rating.
> > Especially, I would like to hear the thoughts of reviewers PQv1, cgm5 on the author response.

---

### Author Response · Authors · 2023-11-20

We would like to thank all reviewers for their time and feedback! We've included much of the feedback already in a revised version of the paper (revision changes are highlighted in red).

Overall, we were encouraged that you think our work tackles an important and timely research question (9JCy, 59VJ, cgm5) in a well-written and easy-to-follow manner (9JCy, 59VJ, cgm5). We are glad you found that our paper provides novel insights (59VJ) and interesting findings (9JCy) based on convincing experimental results (PQv1) that adequately support our claims (cgm5).

We would like to briefly clarify two main points raised in the reviews:

_Section 3.1: Training: Conditions for Fast Convergence of Second-Order Optimization_
In the infinite-width limit (assuming the NTK parametrization) Jacobian stability (eqn 8) is satisfied, i.e. the NN is well-described by its linearization at initialization and thus in the kernel regime. In this lazy regime, NGD converges provably faster than gradient descent. We experimentally evaluate whether Jacobian stability is satisfied for architectures used in practice to test whether these networks are well-described by the infinite-width limit.

_Section 3.3: Continual Learning: Catastrophic Forgetting_
It's been previously observed that catastrophic forgetting can be mitigated by increasing NN width. We corroborate this finding in very wide toy architectures, but demonstrate that in NNs used in practice for computationally feasible widths, the reason for this observation is likely early stopping, not that the NNs behave as if in the kernel regime.

If there are any follow-up questions, we would be happy to answer them before the end of the discussion period.

---

### Meta-Review · Area_Chair_TPKX · 2023-12-09

**Metareview:**

This paper empirically studies the validity of certain theoretical and algorithmic consequences based on the Neural Tangent Kernel (NTK) theory, and shows that some of them fall short in practice: (i) faster convergence of Natural Gradient Descent, (ii) uncertainty quantification in neural contextual bandits, and (iii) catastrophic forgetting in continual learning.

The reviewers found the paper well-written and containing some novel insights. On the other hand, the paper equates the infinite-width limit with the NTK regime, which is not precise because (1) infinite width is not the only way to get the kernel behavior; large initialization could also lead to the NTK regime, and (2) there are other infinite-width parameterizations which leads to non-kernel regimes, such as the mean-field regime and muP parameterization. Equating infinite width the NTK could be misleading to the readers. Furthermore, it has been well established that practical neural networks are not in the NTK regime, as supported by numerous experiments in prior work. This paper looks at different angles, but the main message is somewhat limited given prior work: given that the network is not in the NTK regime, it is not surprising that some of the assumptions made in NTK-based analyses will not be satisfied and some conclusions drawn from there will be flawed.

**Justification For Why Not Higher Score:**

The main message of the paper is somewhat limited given the previous work that already established the disconnect between the NTK theory and practical neural networks.

**Justification For Why Not Lower Score:**

N/A

---

### Decision · Program_Chairs · 2024-01-16

Reject